# Towards Transparent and Explainable Attention Models, ML Reproducibility Challenge 2020

## Reproducibility Summary

**Scope of Reproducibility**

The original paper proposes new LSTM variants that are supposed to improve the transparency and interpretability of the attention mechanism. They claim that modified LSTMs do not hurt the performance (Claim 1) and the attention distributions in the modified model variants provide a faithful explanation of the model's predictions (Claim 2). Additionally, we provide our extensions. We verify whether the mean conicity of the attention values in the Vanilla Transformer is similar to the conicity of the Vanilla LSTM (Claim 4), whether the performance of a modified Transformer does not significantly hurt performance across different datasets (Claim 5) and if the orthogonalization procedure has homogeneous effects across 4 languages on the sentiment classification task (Claim 6).

**Methodology**

We have used the original author's code and the experiments were run on the Lisa cluster with GTX1080Ti GPU. For experiments that extended the original paper's results we have re-implemented part of the existing pipeline and added respective job scripts that were run on the cluster.

**Results**

All the claims stated by the original paper were confirmed. However, not all our experiments overlap with corresponding experiments in the original paper. When reproducing the experiment about POS tags for QQP dataset [1] we did not obtain large cumulative attention to punctuation marks and our results of the rationale experiment are not conclusive. Additionally, we found the Vanilla Transformer model had 3 times smaller conicity values than the Vanilla LSTMs. Constraining the conicity in the transformer's self-attention mechanism did not significantly hurt performance.

**What was easy**

It was easy to run the code published by the authors and in our opinion, overall the paper is very good from a reproducibility point of view.

**What was difficult**

Some of the datasets (MIMIC ICD9 Anemia and MIMIC ICD9 Diabetes [2]) were not easily available due to licensing issues. There were couple of very large datasets (Yelp [3, 4]) and Amazon [3, 5]) that took too much time and computational resources to run (more than 24 hours on GTX1080Ti). It's also important to note that the package runs either on Linux or MacOS with Python 3.7. For better reproducibility, we would recommend the authors to include a table with dataset sizes or their training times.

**Communication with original authors**

We did not have any contact with the original authors.

# 1 Introduction

Attention-based mechanisms [6] have recently gained popularity due to the boost in performance and robustness that they offer across problems in the Natural Language Processing domain. Another convenient feature of those models is that the attention distribution can be visually analyzed to flag words and phrases that trigger the model's decision-making. Although this approach has been widely adopted by the research community [7] as a way to improve the model's explainability, some scientists argue that this method provides neither faithful nor plausible insights [8].

Authors of [9] suggest that this might be due to the lack of variability in the hidden state representations. To overcome this problem, they introduce diversity-driven training and the orthogonalization for LSTMs to increase the variability of the hidden states.

In this work, we attempt to reproduce the results that were carried out in [9] and extend it with multilingual datasets [10]. Additionally, we apply their idea to Transformer models [11] and ensure a reproducible environment by containerizing the software [1] with Docker [12]. We have managed to reproduce most of their results, however, some datasets were not easily available due to the licensing issue, or were too large to process.

# 2 Scope of reproducibility

The original paper introduces an extension to the LSTM models, which makes the model more transparent. The goal is to make the hidden layers correspond to the explainability, so the insights into the model are not only plausible but also faithful. The main claims that we were able to reproduce from the original paper are as follows:

- **Claim 1:** The performance of modified LSTM variants does not significantly (1-3% accuracy change) hurt performance across different tasks and datasets.
- **Claim 2:** The attention distributions in the modified model variants provide a faithful explanation for the model's predictions which is exhibited by low-fraction of hidden state deletions leading to a different outcome (Figure 2) and by large output difference when the attention weights are randomly permuted (Figure 3).

Additionally, we introduce other claims from the extended work on Transformers and multilingual data:

- **Claim 3:** The mean conicity of the attention values in the Vanilla Transformer is similar to the conicity of the Vanilla LSTM.
- **Claim 4:** The performance of a modified Transformer (an extension) does not significantly (1-3%) hurt performance across different datasets.
- **Claim 5:** The orthogonalization procedure has homogeneous effects across 4 languages on the sentiment classification task.

# 3 Methodology

We used the publicly available code provided by the author [2]. We could run experiments without major issues by following instructions in the Readme file [3].

## 3.1 Model descriptions

In [9], there were two proposed LSTM variants - Orthogonal LSTM and Diversity LSTM which were compared to the vanilla LSTM as a baseline.

### 3.1.1 Vanilla LSTM

Depending on task, we use a single or a pair of input, for double version we encode $P = \{w_1^p, ..., w_m^p\}, Q = \{w_1^q, ..., w_n^q\}$ by passing their embedding through LSTM encoder.

$$\begin{aligned} \mathbf{h}_t^p &= LSTM_P\left(e\left(w_t^p\right), \mathbf{h}_{t-1}^p\right) \forall t \in [1, m] \\ \mathbf{h}_t^q &= LSTM_Q\left(e\left(w_t^q\right), \mathbf{h}_{t-1}^q\right) \forall t \in [1, n] \end{aligned}$$

---

[1]Our code repository is available at: `https://anonymous.4open.science/r/1bbc9de7-18c6-4c39-a67e-a4596874ba6b/`

[2]Authors' repository is available at `https://github.com/akashkm99/Interpretable-Attention`

[3]The only minor step missing from the manual is the need to download NLTK POS taggers in the environment by running nltk.download('averaged_perceptron_tagger') and nltk.download('universal_tagset')

With $e(w)$ as embedding for the word $w$. We attend to the intermediate representations of $P, H^P = \{h_1^p, ..., h_m^p\} \in \mathbb{R}^{mxd}$ using the last hidden state $h_n^q \in \mathbb{R}^d$ as the query, using the attention mechanisms.

$$
\begin{aligned}
\tilde{\alpha}_t &= \mathbf{v}^T \tanh\left(\mathbf{W}_1 \mathbf{h}_t^p + \mathbf{W}_2 \mathbf{h}_n^q + \mathbf{b}\right) \forall t \in [1, m] \\
\alpha_t &= softmax\left(\tilde{\alpha}_t\right) \\
\mathbf{c}_\alpha &= \sum_{t=1}^m \alpha_t \mathbf{h}^p t
\end{aligned}
$$

For tasks with a single input sequence, we use a single LSTM to encode the sequence, followed by an attention mechanism (without query) and a final output projection layer.

### 3.1.2 Diversity LSTM

A further extension to the vanilla LTSM model with an additional objective to minimize the conicity, together with maximizing the log-likelihood of the training data, while training the model.

$$
L(\theta) = -p_{model}(y \mid \mathbf{P}, \mathbf{Q}, \theta) + \lambda conicity\left(\mathbf{H}^P\right)
$$

Where $y$ is a ground truth class, $P$ and $Q$ are the input sentences, $H^p = h_1^p, ..., h_m^p \in \mathbb{R}^{m \times d}$ contains all the hidden states of the LSTM, $\theta$ is a collection of the model parameters and $p_{model}(.)$ represents the model output probability. $\lambda$ is a hyperparameter that controls the weight given to diversity in hidden states during training.

### 3.1.3 Orthogonal LSTM

A model similar to vanilla LSTM, except that we replace vanilla LSTM with orthogonal LSTM architecture. To ensure the low conicity, we orthogonalize the hidden states.

$$
\begin{aligned}
\mathbf{f}_t &= \sigma\left(\mathbf{W}_f \mathbf{x}_t + \mathbf{U}_f \mathbf{h}_{t-1} + \mathbf{b}_f\right) \\
\mathbf{i}_t &= \sigma\left(\mathbf{W}_i \mathbf{x}_t + \mathbf{U}_i \mathbf{h}_{t-1} + \mathbf{b}_i\right) \\
\mathbf{o}_t &= \sigma\left(\mathbf{W}_o \mathbf{x}_t + \mathbf{U}_o \mathbf{h}_{t-1} + \mathbf{b}_o\right) \\
\hat{\mathbf{c}}_t &= \tanh\left(\mathbf{W}_c \mathbf{x}_t + \mathbf{U}_c \mathbf{h}_{t-1} + \mathbf{b}_c\right) \\
\mathbf{c}_t &= \mathbf{f}_t \odot \mathbf{c}_{t-1} + \mathbf{i}_t \odot \hat{\mathbf{c}}_t
\end{aligned}
$$

$$
\begin{aligned}
\hat{\mathbf{h}}_t &= \mathbf{o}_t \odot \tanh\left(\mathbf{c}_t\right) \\
\overline{\mathbf{h}}_t &= \sum_{i=1}^{t-1} \mathbf{h}_i \\
\mathbf{h_t} &= \hat{\mathbf{h}}_t - \frac{\hat{\mathbf{h}}_t^T \overline{\mathbf{h}}_t}{\overline{\mathbf{h}}_t^T \overline{\mathbf{h}}_t} \overline{\mathbf{h}}_t
\end{aligned}
$$

where $W_f, W_i, W_o, W_c \in \mathbb{R}^{d_2 \times d_1}, U_f, U_i, U_o, U_c \in \mathbb{R}^{d_2 \times d_1}, b_f, b_i, b_o, b_c \in \mathbb{R}^{d_2}, d_1$ and $d_2$ are the input and hidden dimentions respectively.

### 3.1.4 Extension: Transformer

We further investigate the impact of orthogonalization of attended states on another attention-based model architecture: the transformer [13]. In this paper we focus on classification tasks, containing one input sequence. This section treats the transformer block used in later experiments.

We first describe the self-attention layer. The main ingredient thereof, the dot product attention, takes as input a set of queries $Q \in \mathbb{R}^{T \times k}$, keys $K \in \mathbb{R}^{T \times k}$ and values $V \in \mathbb{R}^{T \times k}$ where $T$ is the sequence length, and $k$ is the hidden dimension of the queries, keys, and values. Given a $k$ dimensional representation of the input tokens and positions $X = \{x_1, ...x_T\} \in \mathbb{R}^{T \times k}$, we define the queries, keys and values as linear transformations of the $i$-th input vector where the transformation matrices are learnable, $k \times k$ parameters. To enable attending to different aspects of the sequence we use a multi-head attention mechanism, where the number of heads is denoted by $h$. Multi-head attention creates $h$ separate attention mechanisms, concatenates them, and learns the interaction between them through a linear layer, parametrized by $W_o \in \mathbb{R}^{h \cdot k \times t}$ that brings the dimension from $T \times h \cdot k$ back to $T \times k$. This can be summarized by the following equations:

$$
Multihead(Q, K, V) = Concat(head_1, ..., head_h)W_O, head_i = Attention(Q^i, K^i, V^i)
$$

$$
Attention(Q, K, V) = softmax\left(\frac{QK^T}{\sqrt{k}}\right)V, Q = XW_q, k = XW_k, V = XW_v
$$

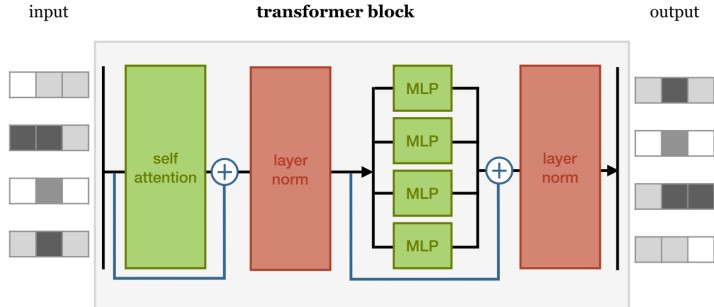

Figure 1: Visualization of a transformer block [11].

Now we relate back to the orthognalization procedure of the paper by Mohankumar et al. [9]. In the self-attention layer described above, for each attention-head, each $x_i$ will have a corresponding attention output, $y_i$ which is a weighted average over the attention values, $v_i$. This can be described by the equation $y_i = \sum_j^T (softmax\left(\frac{QK^T}{\sqrt{k}}\right))_{ij} v_j$, where $V = \{v_1, ...v_T\} \in \mathbb{R}^{T \times k}$. Following the main conjecture of [9], we aim to minimize conicity of representations being attended to, in this case the attention values. In particular, we minimize the average conicity of the values coming from different attention-heads.

Finally, we describe the transformer block, visualized in Figure 1. It consists of (1) self-attention layer, (2) layer normalization over the embedding dimension, (3) a feed-forward layer applied independently to each vector, and (4) another layer normalization. Residual connections are added between the input and (1) as well as between (2) and (3). Finally, the max-pooling operation is applied to the output of the transformer block along the time dimension followed by a linear layer that brings down the dimension to the number of classes.

We investigate two variants of the transformer: a baseline, **Vanilla Transformer** to minimize the negative log-likelihood of the training data. A second variant referred to as **Diversity Transformer** additionally minimizes the mean conicity of the attention values across attention heads.

$$L(\theta) = -p_{model}(y|X, \theta) + \frac{\lambda}{h} \sum_i^h conicity(V^i),$$

where $y$ is the ground truth class, $X$ is the input sequence, $V^i = \{v_1^i, ..., v_T^i\} \in \mathbb{R}^{T \times k}$ contains the attention values of the $i$-th head, $\theta$ is a collection of the model parameters and $p_{model}(.)$ represents the model output probability. $\lambda$ is a hyperparameter that controls the weight given to diversity in attention values during training.

### 3.2 Datasets

#### 3.2.1 Original datasets

The original paper used 12 datasets which are well described by [9] and [14]. Therefore we refrain from repeating their description in this work, but at the same time strongly advise the reader to familiarize himself with the relevant sections before going further.

We managed to reproduce 7 out of these 12 datasets. Two datasets (*Anemia* and *Diabetes*) are part of a larger medical database which is only available after going through an online course and sending an application - due to time constraints we didn't try to obtain them. Another dataset that we didn't use is the *CNN* dataset because the download link expired and as confirmed by the author is no longer available. Finally, we didn't obtain results for the *Yelp* [3, 4] and *Amazon* [3, 5] datasets because they were too computationally too expensive to train. Regarding the datasets that we were able to reproduce, all of them were automatically downloaded using a script as described in the repository apart from *Twitter ADR* dataset which we obtained from authors of [14] via email.

All reproduced text classification datasets fall in the category of binary classification and all of them are almost balanced (less than 1% difference in class distribution). In Table 1 we include a number of examples for training, development, and testing sets of all reproduced datasets.

| split/dataset | 20News | IMDB | Tweets | SST | Babi 1 | Babi 2 | Babi 3 | SNLI | QQP |
|---|---|---|---|---|---|---|---|---|---|
| **train** | 1236 | 17212 | 17003 | 6920 | 8500 | 8500 | 8500 | 549,367 | 327,462 |
| **dev** | 310 | 4304 | - | 872 | 1500 | 1500 | 1500 | 9842 | 36384 |
| **test** | 387 | 4363 | 4251 | 1821 | 1000 | 1000 | 1000 | 9824 | 40430 |

Table 1: Number of examples per split for all reproduced datasets.

### 3.2.2 Cross-language sentiment classification

In this work, we have extended the experiments by including the sentiment classification task on the cross-lingual dataset CLS [4]. This dataset is the most similar to the Amazon dataset that was used in the original paper [9]. The dataset has been already preprocessed to balance the labels by original authors [10]. The dataset consists of 12.000 data samples for each of the 4 languages: English, German, French, and Japanese. The dataset is split into train, test, and dev datasets that consist of 6000, 3000, and 3000 samples respectively. Originally, the dataset was divided into subcategories of reviews (book, movies, and DVD), however, for the sentiment classification we use all categories together. We provide a script available at our code repository that automatically downloads, preprocesses, and tokenizes the data. The preprocessing is carried out as in the original paper's implementation [10]. Firstly, we split words on delimiters, map digits into *<num>* token, remove punctuation, and convert to lower case letters. Additionally, for English data we normalize contraction (e.g. "don't" to "do not"). Finally, we convert corresponding sentiment labels (i.e. "positive", and "negative") to integer representations.

## 4 Results and Discussion

In the following section, we compare our reproduced results to the results shown in the original paper. We briefly discuss whether claims of the original paper are supported by our experiments.

### 4.1 Results reproducing original paper

Table 2 shows that Claim 1 does hold in our results. Our results are similar to the original paper, apart from BAbI 2 and 3 for which the accuracy differs significantly. However, in general, the performance is comparable for all model variants and the conicity is significantly lower for Diversity and Orthogonal LSTM.

| Dataset | LSTM | | Diversity LSTM | | Orthogonal LSTM | |
|---|---|---|---|---|---|---|
| | Accuracy | Conicity | Accuracy | Conicity | Accuracy | Conicity |
| **Our results** | | | | | | |
| SST | 79.83 | 0.77 | 78.14 | 0.19 | **80.0** | 0.29 |
| IMDB | **89.6** | 0.55 | 88.29 | 0.08 | 87.21 | 0.15 |
| 20News | **93.28** | 0.77 | 92.44 | 0.13 | 92.44 | 0.22 |
| Tweets | 82.61 | 0.78 | **84.99** | 0.26 | 83.99 | 0.24 |
| SNLI | **78.14** | 0.59 | 73.66 | 0.04 | 76.86 | 0.29 |
| QQP | 78.49 | 0.57 | 78.16 | 0.03 | **78.56** | 0.32 |
| bAbI 1 | **100.0** | 0.72 | 99.8 | 0.06 | **100.0** | 0.22 |
| bAbi 2 | 57.9 | 0.52 | 48.9 | 0.12 | **61.1** | 0.19 |
| bAbi 3 | 17.4 | 0.99 | 41.3 | 0.06 | **57.9** | 0.16 |
| **Original results** | | | | | | |
| SST | **81.79** | 0.68 | 79.57 | 0.20 | 79.54 | 0.28 |
| IMDB | **89.49** | 0.69 | 88.54 | 0.08 | 88.71 | 0.18 |
| 20News | **93.55** | 0.77 | 91.03 | 0.15 | 92.15 | 0.23 |
| Tweets | 87.02 | 0.77 | **87.04** | 0.24 | 83.20 | 0.27 |
| SNLI | **78.23** | 0.56 | 74.16 | 0.04 | 76.46 | 0.27 |
| QQP | **78.74** | 0.59 | 78.40 | 0.04 | 78.61 | 0.33 |
| bAbI 1 | 99.10 | 0.56 | **100.00** | 0.07 | 99.90 | 0.22 |
| bAbI 2 | 40.10 | 0.48 | 40.20 | 0.05 | **56.10** | 0.21 |
| bAbI 3 | 47.70 | 0.43 | 50.90 | 0.10 | **51.20** | 0.12 |

Table 2: Accuracy and conicity of Vanilla, Diversity and Orthogonal LSTM across different datasets.

In Figure 2 we check what is the fraction of removed attention weights required to change the prediction. In theory, a low fraction removed before decision flip confirms the faithfulness of the model. We observe that our and the original results are almost identical which confirms Claim 2. Interestingly, the Tweets dataset requires a larger fraction of hidden

---

[4]Dataset is available at `https://zenodo.org/record/3251672#.YAll0-hKibg`

states to be deleted than sentiment classification datasets. That is perfectly reasonable as classifying whether a tweet contains adverse drug reaction is more nuanced than sentiment analysis and requires attention to a larger fraction of the input sentence. SNLI presents similar behavior to QQP as both tasks concern relationships between a pair of input sentences.

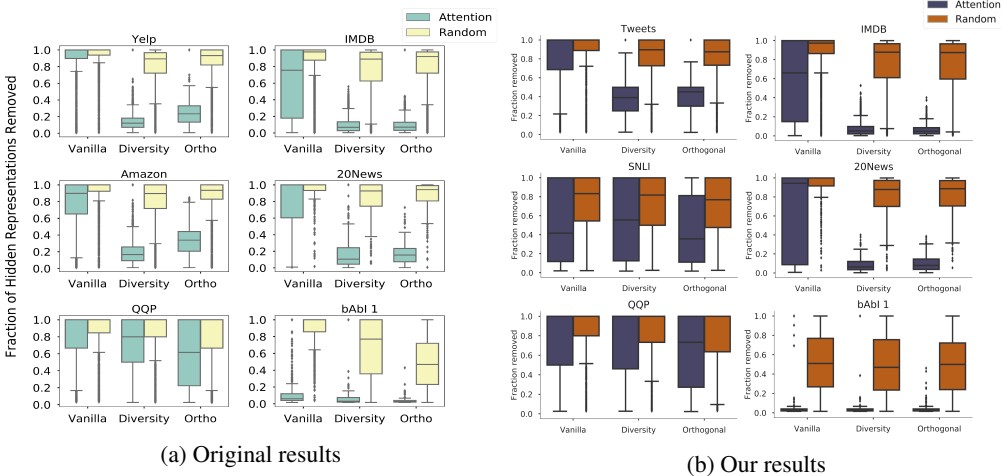

(a) Original results

(b) Our results

Figure 2: Box plots of fraction of hidden representations removed for a decision flip. Dataset and models are mentioned at the top and bottom of figures. Blue and Yellow indicate the attention and random ranking

The next experiment checks how random permuting attention weights affects predicted outputs. Supposing a model is faithful, permutation should lead to a large output difference. Figure 3 is the second argument supporting Claim 2 for binary classification. Nevertheless, once again we see that QQP and SNLI behave differently and for those datasets, attention permutation isn't that detrimental for Diversity and Orthogonal LSTM. This can be explained by the fact that for sentence pair tasks the attention weights are more homogenous therefore the permutation has a smaller effect.

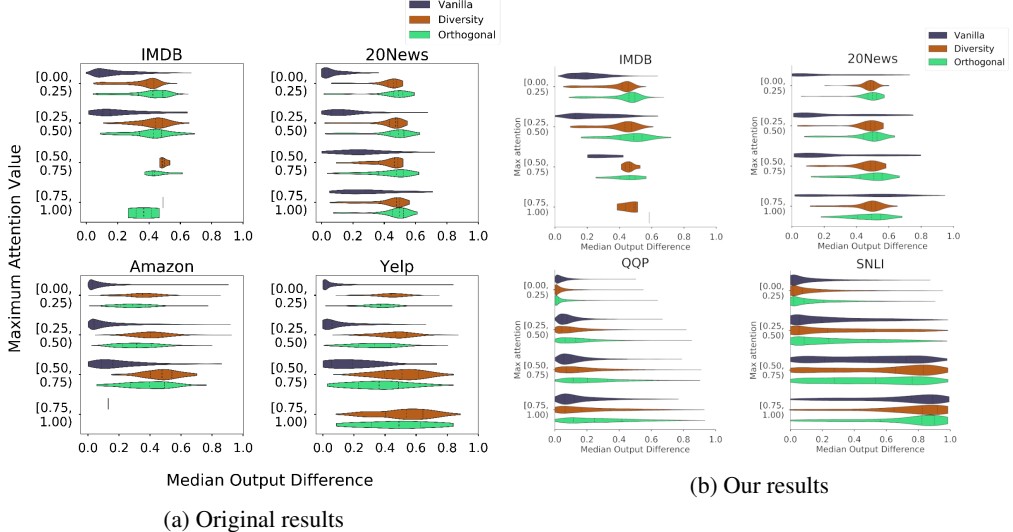

(a) Original results

(b) Our results

Figure 3: Comparison of Median output difference on randomly permuting the attention weights in the vanilla, Diversity and Orthogonal LSTM models. The Dataset names are mentioned at the top of each figure.Colors indicate the different models as shown legend.

Figure 4 verifies the plausibility assumption i.e. whether words with high attention intuitively seem important for the correct prediction. Here the authors focus on Part of Speech tags and the cumulative attention each tag receives. They claim that for Yelp, Amazon, and QQP datasets vanilla LSTM assigns disproportionally high attention to punctuation tags which are meaningless in tasks such as sentiment classification. While we weren't able to reproduce Yelp and Amazon datasets due to their size, our results for QQP do not show large attention attribution to punctuation. Therefore,

| Dataset | Vanilla LSTM | | Diversity LSTM | | Orthogonal LSTM | |
|---|---|---|---|---|---|---|
| | Rationale Attention | Rationale Length | Rationale Attention | Rationale Length | Rationale Attention | Rationale Length |
| **Our results** | | | | | | |
| SST | 0.827 | 0.796 | 0.652 | 0.148 | 0.518 | 0.183 |
| IMDB | 0.885 | 0.812 | 0.903 | 0.208 | 0.939 | 0.356 |
| 20News | 0.936 | 0.733 | 0.919 | 0.153 | 0.859 | 0.201 |
| Tweets | 0.861 | 0.833 | 0.502 | 0.244 | 0.466 | 0.326 |
| **Original results** | | | | | | |
| SST | 0.348 | 0.240 | 0.624 | 0.175 | | |
| IMDB | 0.472 | 0.217 | 0.761 | 0.169 | | |
| 20News | 0.627 | 0.215 | 0.884 | 0.173 | | |
| Tweets | 0.284 | 0.225 | 0.764 | 0.306 | | |

Table 3: Mean Attention given to the generated rationales with their mean lengths (in fraction)

it seems that this observation which occurred in 2 out of 12 datasets is more of an artifact of these two datasets, rather than a general problem. That is not to say that the vanilla model has plausible explanations, but rather that this particular experiment cannot properly verify this claim.

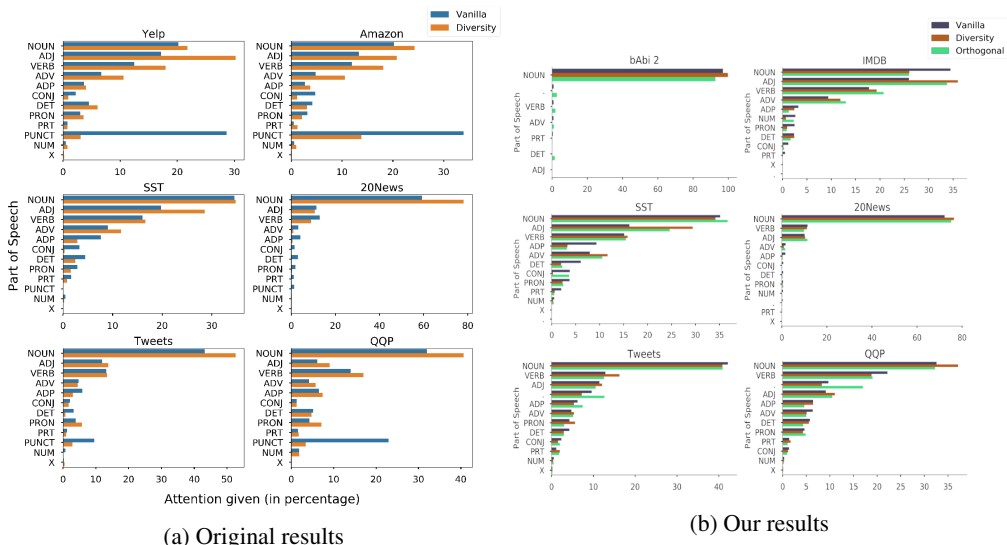

(a) Original results

(b) Our results

Figure 4: Distribution of cumulative attention given to different part-of-speech tags in the test dataset. Blue and Orange indicate the vanilla and Diversity LSTMs.

Finally, we consider rationale which is a minimal set of tokens required to make the correct prediction. The *rationale length* is the fraction of tokens included in a rationale out of all sentence tokens and *rationale attention* is cumulative attention of the words that are included in the rationale out of all words in the input sentence. High rationale attention and low rationale length mean that the model focuses on a couple of important words. Therefore rationale attention is positively correlated with rationale length - the more words are in the rationale, the more cumulative attention is attributed. We see in Table 3 that the cumulative rationale attention of vanilla LSTM in our results is similar or sometimes higher than Diversity LSTM and much higher than vanilla LSTM in the original results. This speaks in favor of denying Claim 2. However, the length is also much higher so it might be just a matter of the hyperparameter determining rationale length which was used for generating the rationale. That's why this experiment neither confirms nor denies this claim and would require further research to draw a definitive conclusion.

## 4.2 Multilingual Dataset

Table 5 shows the results for 4 different languages and 3 LSTM models averaged over 5 runs. The results show a similar trend as in the original paper [9]. The best performance is achieved for vanilla LSTM models, however, the conicity values are a few times bigger than for the Diversity and Orthogonal variants. There is a 3-5% decrease in accuracy for Diversity models and 1-3% for Orthogonal models when compared to the vanilla LSTM. The conicity values are very similar between Diversity and Orthogonal LSTMs which might suggest that the latter one achieves a better performance while maintaining a good level of explainability. Also, it is worth noting that original authors achieved much higher

| Dataset | Vanilla Transformer | | Diversity Transformer | |
|---|---|---|---|---|
| | Accuracy | Conicity | Accuracy | Conicity |
| **Binary Classification** | | | | |
| CLS English | **78.09** ± 1.14 | 0.21 ± 0.01 | 77.69 ± 1.11 | 0.11 ± 0.00 |
| CLS German | **77.49** ± 2.00 | 0.21 ± 0.01 | 76.92 ± 0.77 | 0.11 ± 0.00 |
| CLS French | **80.20** ± 1.01 | 0.21 ± 0.00 | 80.11 ± 1.40 | 0.11 ± 0.00 |
| CLS Japanese | 71.54 ± 5.47 | 0.22 ± 0.01 | **75.53** ± 1.78 | 0.11 ± 0.00 |
| SST | **71.25** ± 1.38 | 0.24 ± 0.00 | 70.84 ± 2.18 | 0.13 ± 0.00 |
| IMDB | 80.90 ± 1.26 | 0.24 ± 0.01 | **81.48** ± 0.46 | 0.08 ± 0.00 |
| 20News_sports | 76.02 ± 1.69 | 0.18 ± 0.01 | **78.71** ± 2.52 | 0.12 ± 0.00 |
| Tweets | **89.89** ± 1.20 | 0.25 ± 0.01 | 89.46 ± 0.53 | 0.11 ± 0.00 |

Table 4: Accuracy and conicity of Vanilla and Diversity Transformer across different datasets. Each score represents a mean value over 5 runs with a respective standard deviation.

| Dataset | LSTM | | Diversity LSTM | | Orthogonal LSTM | |
|---|---|---|---|---|---|---|
| | Accuracy | Conicity | Accuracy | Conicity | Accuracy | Conicity |
| **Binary Classification** | | | | | | |
| CLS English | **79.25** ±0.79 | 0.43 ±0.03 | 75.45 ±0.66 | 0.15 ±0.004 | 76.81 ±0.92 | 0.18 ±0.004 |
| CLS German | **78.97** ±2.09 | 0.43 ±0.03 | 74.64 ±1.34 | 0.14 ±0.003 | 77.09 ±1.07 | 0.17 ±0.004 |
| CLS French | **81.60** ±0.60 | 0.46 ±0.02 | 77.61 ±1.23 | 0.14 ±0.003 | 78.93 ±0.88 | 0.17 ±0.006 |
| CLS Japanese | **78.87** ±0.80 | 0.44 ±0.03 | 75.78 ±1.35 | 0.15 ±0.004 | 77.05 ±0.70 | 0.18 ±0.010 |

Table 5: Results for cross-lingual sentiment classification task run on Amazon reviews for 4 languages. Each score represents a mean value over 5 runs with a respective standard deviation.

accuracy scores on the Amazon dataset (the best accuracy is 93.73%). This is because we don't use any pre-trained word embeddings during the preprocessing steps due to inconsistency between languages.

### 4.3 Transformers

Table 4 portrays the accuracies and mean conicity of the attention values for various datasets. The purpose of this experiment is to verify if the transformer's self-attention mechanism also tends to attending to highly similar token representations which in the original paper [9] caused the attention weights to carry little meaning. We find that, at least for classification tasks, the conicity of the attention values in the Vanilla Transformer is significantly smaller (mean 0.24) compared to the conicity of the hidden states (mean 0.58) in the Vanilla LSTM's. This leads to the rejection of Claim 3, as these values differ significantly. This result suggests that the problem of the attention weights failing to offer a plausible explanation of the prediction might be less severe in the Vanilla Transformers as compared to the Vanilla LSTMs. However, subsequent experiments would have to be conducted to confirm this statement.

Similarly to the findings from Section 4.1, we find that the performance of diversity-driven training does not significantly differ from the performance of vanilla transformer (mean difference 1.15), which confirms Claim 4.

## 5 Conclusions

The majority of the experiments from the paper were reproducible. In the reproduced results, we have also noticed that modified LSTM variants do not significantly degrade the performance. Additionally, our results seem to support the claim that the LSTM variants provide more faithful explanations for the model's predictions.

Going a step further, the experiments were extended by adding multilingual datasets and repeating part of the experiments on the Transformer model. Interestingly, a similar performance drop across LSTM variants could be observed for multilingual datasets. The vanilla Transformer model had much smaller conicity values than the vanilla LSTMs. However, the diversity-driven training does not impact the performance of Transformer variants.

Overall, we are quite pleased with obtained results and the quality of original paper's implementation.

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

# Appendix

## 1 Hyperparameters

### 1.1 Experiments with LSTM-based model

We used the default hyperparameters provided by the authors as we assumed that those were used for the presented results. For all experiments, we used a single-layer LSTM and the Adam [1] optimizer with a learning rate of 0.001 and weight decay 0.00001. Hyperparameters that are variable between datasets are shown in Table 1. For a given dataset, the same hyperparameters were used for all three model variants, except that Diversity LSTM had one additional parameter - diversity weight - which was set to 0.5.

| Dataset | Batch size | Embedding size | Hidden size | # epochs | Vocabulary size |
|---------|-----------|----------------|-------------|----------|-----------------|
| SST | 32 | 300 | 128 | 8 | 13826 |
| IMDB | 32 | 300 | 128 | 8 | 12487 |
| 20News | 32 | 300 | 128 | 8 | 6515 |
| bAbI 1 | 50 | 50 | 32 | 100 | 24 |
| bAbi 2 | 50 | 50 | 64 | 200 | 38 |

Table 1: Dataset specific hyperparameters

### 1.2 Experiments with multilingual datasets

| Dataset | Batch size | # epochs | Vocabulary size |
|---------|-----------|----------|-----------------|
| CLS English | 32 | 4 | 2128 |
| CLS German | 32 | 4 | 1830 |
| CLS French | 32 | 4 | 1457 |
| CLS Japanese | 32 | 4 | 1767 |

Table 2: Dataset specific hyperparameters for multilingual datasets

### 1.3 Experiments with Transformer based models

For all experiments, we used a single transformer block. We used the same pre-trained word embeddings as done in the original paper. For the multilingual dataset, we learnt the embedding size of 300. Additionally, we learnt the positional embeddings with the dimensionality of 125% of the longest input in the training set or of the size specified by the authors. The size of the hidden layers of the feed-forward component of the transformer block was 4 times as big as the input and output. We used an Adam optimizer with a learning rate of 0.0001 and a weight decay of 0.00001. Each model was trained for 15 epoch with batch size 16. The model with the best validation accuracy was then evaluated on the test set. Train, validation and test split are copied from the original paper. The diversity weight for the Diversity Transformer was set to 0.5 for all experiments.

## References

[1] Diederik P Kingma and Jimmy Ba. Adam: A method for stochastic optimization. *arXiv preprint arXiv:1412.6980*, 2014.

