# OpenReview forum: "Towards Transparent and Explainable Attention Models, ML Reproducibility Challenge 2020"
_ML_Reproducibility_Challenge/2020 — Reject_

### Official Review · AnonReviewer1 · 2021-02-28
**Review of "Towards Transparent and Explainable Attention Models, ML Reproducibility Challenge 2020"**

**Rating:** 8
**Confidence:** 4

**Review:**

This is a well-written and faithful replication of the original work. The authors describe the methods well and do their best to reproduce and extend studies of the original methods to multilingual settings, providing their own code along the way. Though some of the original datasets were not accessible to the authors, they swapped these out for extended studies which seem to show consistent results.

Overall, I think the paper was well-written and met the expectations of a great reproduction. Perhaps, some additional discussion on the intuition behind adding multilingual experiments and the potential connection to conicity might be worthwhile to motivate the additional experiments a bit more. In general, additional discussion of findings might have been helpful to the reader, but not necessary. The description of the original method served as a nice summary of the original work as well.

Typos:

In the conclusion: "Majority of..." -> "The majority"

Appendix 1, citation for adam is broken.

**Familiar With The Original Paper:**

I have not read the original paper

**Reproducibility Summary:**

Report has summary

---

### Official Review · AnonReviewer2 · 2021-03-01
**Reproduces most results from the original paper and extends to own experiments**

**Rating:** 7
**Confidence:** 4

**Review:**

 I can confirm that the authors included a clear and concise reproducibility summary, noting that they reused some of the original code and data and made a few modifications to original code base to extend the experiments. The authors highlight that there were some challenges working with the data where licensing was involved and in other cases where the datasets were large.  The authors do not try new hyper-parameters, but re-use the hyper-parameters from the original paper. Ablation or recommendations to the original authors are not provided, however the authors do provide some results for independent experimentation on the CLS dataset. The paper is otherwise well written and easy to read.

**Familiar With The Original Paper:**

I have read the original paper

**Reproducibility Summary:**

Report has summary

---

### Official Review · AnonReviewer3 · 2021-03-08
**Great report, good set of additional ablation results. Code is not accessible.**

**Rating:** 6
**Confidence:** 5

**Review:**

* Reproducibility Summary

  The report contains a well-defined and articulate reproducibility summary as prescribed by the challenge.
* Scope of reproducibility

  The report contains well-defined scope involving six central claims of the original paper. The paper also investigates additional claims on Transformers and multilingual data.
* Code: whether reproduced from scratch or re-used author repository.

  Authors provide their own codebase link. However, I was unable to open it - seems the link https://github.com/KacperKubara/Transparency does not exist. Neither did I find any code in the appendix. I'll be happy to increase my score if this is fixed.
* Communication with original authors

  The report mentions they did not communicate their results/findings to the original authors.
* Hyperparameter Search

  The authors did not perform any additional hyperparam search, they seem to have only run the default ones provided by the author. This is a missed opportunity in my view, as the authors could have explored various hyperparam choices to see if the results hold more robustly.
* Ablation Study

  The authors perform an ablation study by introducing Transformers and multi-lingual data. This kind of ablation study is perfect and highly appreciated for a reproducibility report.
  It's interesting to find the conicity of Transformers to be much smaller. However, that doesn't totally imply Vanilla Transformers to have a more plausible explanation of attention weights. Also, the choice of evaluating the same on Transformers is tricky, as multi-head attention systems it is generally difficult to pinpoint the attention contribution of a single word.  The authors could have compared their results with that of what BERT looks at https://arxiv.org/abs/1906.04341, but in any case, this additional set of experiments are quite welcome.

* Discussion on results

  The report provides information on which parts of reproduction are easy and which are difficult. Not surprisingly, certain datasets are difficult to procure and some are harder to train. The report does a good job mentioning these.
* Recommendations for reproducibility

  The authors highly commend the original paper on their state of reproducibility.
* Overall organization and clarity

  Overall, minor typos but not that significant. In Section 4.1 the authors might have forgotten to comment out the line "Logically group related results into sections"


**Familiar With The Original Paper:**

I have read the original paper

**Reproducibility Summary:**

Report has summary

---

### Decision · Program_Chairs · 2021-03-31

**Decision:**

Reject

**Comment:**

Overall reviews and/or the paper content not good enough for the AC to recommend to the journal.